# Oil Penetration of Batter-Breaded Fish Nuggets during Deep-Fat Frying: Effect of Frying Oils

**DOI:** 10.3390/foods11213369

**Published:** 2022-10-26

**Authors:** Lulu Cui, Jiwang Chen, Jinling Zhai, Lijuan Peng, Douglas G. Hayes

**Affiliations:** 1College of Food Science and Engineering, Wuhan Polytechnic University, Wuhan 430023, China; 2Key Laboratory for Deep Processing of Major Grain and Oil (Wuhan Polytechnic University), Ministry of Education, Wuhan 430023, China; 3Department of Biosystems Engineering and Soil Science, University of Tennessee, Knoxville, TN 37996, USA

**Keywords:** batter-breaded fish nuggets, deep-fat frying, oil penetration, dielectric constant, viscosity

## Abstract

Four frying oils (rapeseed, soybean, rice bran, and palm oils) were employed either as received (fresh) or after preheating at 180 °C for 10 h, and measured for their fatty acid composition, viscosity, and dielectric constant. Batter-breaded fish nuggets (BBFNs) were fried at 180 °C (60 s), and the effect of the oils’ quality on the oil penetration of fried BBFNs were investigated via the analysis of the absorption and the distribution of fat. Preheating increased the viscosity and dielectric constant of the oils. The total fat content using fresh oils was the greatest for palm oil (14.2%), followed by rice bran oil (12.2%), rapeseed oil (12.1%), and soybean oil (11.3%), a trend that was nearly consistent with the penetrated surface oil, except that the penetrated oil for soybean oil (6.8%) was higher than rapeseed oil (6.3%). The BBFNs which were fried using fresh oils possessed a more compact crust and smaller pores for the core and underwent a lower oil penetration compared to the preheated oils. The results suggested that the oils’ quality significantly affected the oil penetration of fried BBFNs.

## 1. Introduction

Deep-fat frying is a popular way to cook foods by immersing them into frying oil at 150–200 °C [1,2]. Due to the golden-yellow and crispy crust that forms during deep-fat frying, fried batter-breaded foods have a juicy core, which are widely popular food commodities. In recent years, freshwater fish is commonly favored by consumers owing to its high nutritional value and low price, accounting for 63.6% of the aquaculture yield of China in 2020, thereby increasing the fishery yield value of freshwater fish in China [3]. Fried batter-breaded fish nuggets (BBFNs), due to their unique flavor and ease of preparation, are widely consumed in Chinese homes and restaurants. At present, they are produced and sold by many well-known food enterprises, and the market demand shows a rapid growth trend [3]. However, the fat content of batter-breaded foods is generally high, reaching one-third by weight and in some cases up to 50% [4,5]. As a result, the long-term consumption of fried BBFNs may lead to health problems such as cardiovascular disease and hypertension [6,7,8]. Moreover, the oxidation of fatty acids in frying oils also produces very small amounts of dozens of new polar compounds such as aldehydes, ketones, and alcohols, which have a certain potential threat to the health of consumers in sufficient quantities [9,10,11]. Therefore, it is vital to reduce the fat absorption of BBFNs during deep-fat frying.

Three main mechanisms of fat absorption have been hypothesized by researchers: water replacement, the cooling-phase effect, and surfactant theory [4,5,12]. Water replacement theory refers to fat absorption through large pores caused by moisture evaporation during deep-fat frying. The theory of the cooling-phase effect describes a “vacuum effect”: when fried foods are removed from the fryer, the pressure inside the fried foods decreases rapidly, leading to the penetration of oil on the surface into the food’s interior. The fat absorption described by this mechanism represents two-thirds of the final fat content, which is restricted to the microstructure of crust and the viscosity of frying oils [1,12,13]. The surfactant theory pertains to the degradation of frying oils (triglycerides rich in unsaturated fatty acids), resulting in the formation of polar compounds and polymers that exhibit surface activity and reduce the interfacial tension and the contact angle between the oil and water, thereby increasing the fat absorption [12,14,15]. Overall, the quality of frying oils becomes the key factor which affects the fat absorption.

During deep-fat frying, frying oils will undergo a series of complex reactions such as hydrolysis, oxidation, isomerization, and polymerization. Various volatile compounds, monomers, and polymerization products are formed, which change the physicochemical properties of the frying oils, resulting in a quality deterioration of the frying oils [11,16,17]. The degree of unsaturation and the frying time for frying oils will affect the fat absorption [11,18]. When unsaturated fatty acids degrade during deep-fat frying, the contents of the polar compounds and polymers witness a linear increase with time. Meanwhile, as the frying time is prolonged, the degree of polymerization also increases, thereby leading to an increase in the frying oils’ viscosity and dielectric constant, which increases the fat absorption [4,19,20,21].

In our previous studies, we demonstrated that the batter’s xanthan gum/soybean fiber ratio, wheat starch/wheat protein ratio, and the amount of fermented bamboo shoot dietary fiber had a significant effect on the fat content of fried BBFNs [22,23,24]. To the authors’ knowledge, there are no published reports on the effect of a frying oils’ quality on the oil penetration of fried BBFNs. Therefore, in this paper, four oils differing in their degree of unsaturation were used for frying: the oils from rapeseed, soybean, rice bran, and palm were used, which are the most popular among fried foods in both fast food restaurants and food enterprises around the world [10]. Thermally deteriorated frying oils with a high viscosity and dielectric constant were prepared from fresh oils by preheating at 180 °C for 10 h, and then measured for their fatty acid composition, viscosity, and dielectric constant. The fried BBFNs were evaluated for the moisture and fat contents, color, microstructure, and fat distribution to interpret the effect of the frying oils’ quality on the oil penetration of BBFNs during deep-fat frying.

## 2. Materials and Methods

### 2.1. Materials

#### 2.1.1. Organic/Biological Materials

Grass carp (about 1.5 kg) and green tea were purchased from a local supermarket in Wuhan, China. Plain wheat flour was purchased from Wuhan Taiyanghang Food Co., Ltd. (Wuhan, China). Corn starch (starch content 90.1%, dw) was provided by Shandong Jincheng Co., Ltd. (Zhaoyuan, China). Breadcrumbs of a <2 mm particle size were provided by Wuxi Jinhuanghua Food Co., Ltd. (Wuxi, China). Baking powder was obtained from Angel Yeast Co., Ltd. (Yichang, China). The oils from soybean, rapeseed, rice bran, and palm were provided by Yihai Jiali Grain and Oil Industry Co., Ltd. (Wuhan, China). Low-quality frying oils were prepared from the fresh oils by preheating at 180 °C for 10 h.

#### 2.1.2. Chemicals

Boron trifluoride (BF_3_, 14.0% *w*/*v* dissolved in MeOH) was provided by Shanghai Macklin Biochemical Co., Ltd. (Shanghai, China). Sudan Red B was purchased from Shanghai Hengdailao Biological Co., Ltd. (Shanghai, China). Nile Red was obtained from Sigma-Aldrich (St. Louis, MO, USA).

### 2.2. Fatty Acid Composition, Viscosity, and Dielectric Constant of Frying Oils

#### 2.2.1. Fatty Acid Composition

The fatty acid composition of frying oils was determined according to Gao et al. [25] and Kim et al. [26], with slight modifications. The fatty acids were converted to fatty acid methyl esters (FAMEs) before analysis. Frying oil (65 mg) was added into a vessel containing 2 mL of 0.5 M NaOH in methanol. The vessel was heated and shaken at 65 °C for 30 min and then cooled to room temperature (25 °C). After that, two milliliters of the methanolic BF_3_ solution were added into the vessel, which was then heated and shaken at 70 °C for methylation. After 5 min of the reaction, the mixture was cooled to room temperature (25 °C) and hexane (2 mL) and saturated NaCl solution (2 mL) were separately added to the reaction mixture, followed by a thorough shaking. The resultant hexane layer (upper layer) was recovered and then used for gas chromatography-mass spectroscopic (GC-MS) analysis.

The fatty acid composition of frying oils was investigated by using the 7890 NGC/5975 MSD GC-MS instrument from Agilent Technologies Co. (Santa Clara, CA, USA) equipped with a HP-88 capillary column from Agilent (100 m × 0.25 mm × 0.2 μm). Helium was used as a carrier gas and the flow rate was 1 mL/min. The inlet temperature was 280 °C and the injection volume was 1.0 μL with a split ratio of 50:1. The column temperature program was as follows: the temperature was set at 140 °C and held for 5 min, and then raised to 180 °C (10 °C/min), held at 180 °C for 10 min, finally increased to 230 °C (3 °C/min), and held at 230 °C for 1 min. For the detector, the transfer line and MS source temperatures were 250 °C and 230 °C, respectively. The ionization energy of the electron ionization system was set at 70 eV, and the mass range was set at 20 to 500 m/z. The relative percentages of the fatty acid composition were expressed as the direct percentage calculation from the total peak area of all the fatty acids m/z fragments in the fresh oils, without a previous calibration using fatty acid standards.

#### 2.2.2. Viscosity

The viscosity of the frying oils was determined by using the NDJ-7 viscometer (Shanghai Jingke Tianmei Trading Co., Ltd., Shanghai, China) according to Sun et al. [16], with slight modifications. The frying oil was poured into the No. Ⅱ thermostat cup to cover the rotor over 1 mm. The viscometer was opened after the test temperature reached 50 °C. The data was recorded after the rotor was stable. In addition, the thermostat cup was washed and dried between the measurements. The viscosity of the frying oils (η; mPa·s) was calculated according to Equation (1).
H = kA(1)
where k and A represent the coefficient of the rotor and the reading provided by the viscometer, respectively.

#### 2.2.3. Dielectric Constant

The dielectric constant serves as a simple and rapid method determining the polar component content of the frying oils [27]. The frying oil (500 mL) was placed in a beaker. A self-made capacitor was completely immersed in the beaker and connected to the high and low output ends of the JK2817B LCR digital bridge (Changzhou Jin Ai Lian Electronic Technology Co., Ltd., Changzhou, China) with 1 V of test voltage and 1 kHz of frequency. The dielectric constant of the frying oils (*ε*) was calculated according to Equation (2) [28].
(2)$$\varepsilon =\frac{c}{c_{0}}$$
where *c* and *c*_0_ represent the full dielectric capacitance and the vacuum capacitance, respectively.

### 2.3. Preparation of BBFNs

After its purchase at a nearby supermarket, fresh grass carp was immediately transported to the laboratory using foam boxes filled with crushed ice, and then was processed by removing the head, tail, fin, scales, skin, and internal organs. It was cleaned and cut into the nuggets of 6 cm × 1.5 cm × 1.5 cm with a stainless-steel knife, then trimmed to obtain the pieces that were uniform in size (4 cm × 1.5 cm × 1.5 cm; ~10 g). A boiled green tea solution (green tea: deionized water 1:50 *w*/*w*) with NaCl (3%, *w*/*w*) was used to deodorize the trimmed fish nuggets. The ratio of fish to tea solution was 1:1, *w*/*w*. The fish nuggets were allowed to marinate in the tea solution for 0.5 h, and cooled for 15 min.

Corn starch (40 g), plain wheat flour (60 g), baking powder (1.0 g), NaCl (2.0 g), whey protein powder (4.0 g), and deionized water (98 g) were mixed in a blender (RW20. n, IKA Co., Breisgau, Germany) at 2000 rpm to prepare the batter. The fish nuggets were then immersed in the batter for 10 s and allowed to drain for 15 s. The process was repeated until the liquid drainage was absent and followed by the covering of the breadcrumbs. Finally, the BBFNs were put into a blast drying oven (101-BS, Shanghai Yuejin Medical Equipment Co., Ltd., Shanghai, China) at 40 °C for 6 h, and cooled for 30 min.

### 2.4. Frying Process

The BBFNs were fried at 180 °C for 60 s in the fryer (YZ-3032-BC, Guangdong Youtian Household Appliances Co., Ltd., Guangzhou, China). The quality attributes were measured after the fried BBFNs were cooled at 25 °C for 1 h in a stainless-steel strainer to drain the excess oil.

### 2.5. Analysis of Moisture Content

The moisture contents of the crust and core of fried BBFNs were measured according to the NO.922.06 method [29]. The crust and core of the fried BBFNs were separated, chopped, mixed, and sampled evenly. The specimen of the crust and core of the fried BBFNs (5.0 g) were placed in an aluminum box. After being capped, the crust and core were dried to a constant weight in an oven at 101–105 °C.

### 2.6. Analysis of Fat Content

The fat content of the fried BBFNs was measured according to the method given in the published report [30,31].

#### 2.6.1. Standard Curve

Sudan Red B with the frying oil solution (c) (0.4, 0.5, 0.6, 0.7, 0.8 g/L) was diluted with petroleum ether (PE) by 50-fold (*v*/*v*) and its absorbance values (A) were measured at 510 nm using a spectrophotometer. The standard curve was made with a concentration (c) as the ordinate and absorption value (A) as abscissa.

#### 2.6.2. Surface Oil

The BBFNs were fried using 0.4 g/L of Sudan Red B with the oil solution according to the aforementioned frying procedure (Section 2.4). The fried BBFNs were put into a 250 mL beaker with 150 mL of PE, and rinsed for 10 s. The resultant solution was transferred to a 250 mL round-bottom flask, which had been previously dried and weighed (*m*_2_). The extracted oil was collected by evaporating the solvent to a constant mass (*m_1_*). Finally, the amount of surface oil (*m*) was calculated using Equation (3).
(3)$$m=m_{1}-m_{2}$$
where *m*_1_ and *m*_2_ represent the mass of the round-bottom flask with the surface oil (g) and the empty round-bottom flask (g), respectively.

#### 2.6.3. Penetrated Surface Oil

The fried BBFNs which removed the surface oil, as described above, were cut into small pieces and placed into the filter paper tube, and the extraction was carried out for 4 h using a Soxhlet apparatus at 60 °C. The flasks containing the extracted oil were then dried to a constant mass at 105 °C. For analysis, the Soxhlet-extracted oil (*m*_3_) was diluted 50-fold (*v*/*v*) with PE and the absorbance was measured at 510 nm. The corresponding Sudan Red B concentration in the oil (*c*_1_) was calculated according to a standard curve. Consequently, the amount of penetrated surface oil (*m*_4_) was calculated using Equation (4).
(4)$$m_{4}=\frac{m_{3}c_{1}}{0.4\;g/L}$$
where *m*_4_ and *m*_3_ represent the mass of the penetrated surface oil (g) and of Soxhlet-extracted oil (g), respectively, and *c*_1_ represents the Sudan Red B concentration in the oil (g/L).

### 2.7. Instrumental Surface Color Analysis

The surface color of the fried BBFNs was measured using a colorimeter (CR-10, Konica Minolta Holdings, Inc. Tokyo, Japan) according to Chen et al. [22], with slight modifications. Before the test, the colorimeter was preheated for 30 min and calibrated, then the fried BBFNs were placed at the lens mouth of the colorimeter to determine the *L**, *a*,* and *b** values, which represent the brightness (0–100; the greater the value, the greater the brightness), redness, and yellowness, respectively. The positive values of *a** and *b** represent red and yellow, respectively, while the negative values of *a** and *b** designate the green and blue, respectively [32].

### 2.8. Scanning Electron Microscopic (SEM) Analysis

The microstructure of the fried BBFNs was evaluated according to the method described by Zeng et al. [24], with slight modifications. The fried BBFNs from the junction between the crust and core were cut into slices of 2 mm × 2 mm × 1 mm and were handled by using the CO_2_ critical point drying method. The slices were first fixed for 3 h using 2.5% glutaraldehyde prepared with 0.1 mol/L of phosphate-buffered saline (PBS). The fixed slices were washed with 0.1 mol/L of PBS three times, and each washing time was 15 min. Secondly, the dehydration process was sequentially performed with 30%, 50%, 70%, 90%, and 100% ethanol solution for 10 min, respectively. Finally, anhydrous ethanol and isoamyl acetate were used to replace the ethanol. The dehydration time was 10 min. The surface structure of the fried BBFNs slices was assessed by scanning electron microscopy using the S-3000N instrument from Hitachi Co., Ltd., (Tokyo, Japan) with an accelerating voltage of 15 kV and a magnification of 500×. Before the observation, the electron microscope carrier was gilded by high-temperature carbon spraying in a gold-plating apparatus.

### 2.9. Oil Transport via Optical Microscopy

The dyed oil was prepared by dissolving 0.75 g of Sudan Red B into 1.5 L of frying oil. The dye-frying oil solution was heated at 60 °C for 4 h to obtain a uniform solution [14]. The BBFNs were fried in the dyed oil according to the aforementioned frying procedure (Section 2.4). After being cooled at 25 °C, the fried BBFNs were cut into thin slices (5 mm × 3 mm × 3 mm) from the junction between the crust and core with a stainless-steel knife. The oil transport was observed across the cross section of the transversely dissected BBFNs by the optical microscope (XSP-BM-4C, Shanghai BM Optical Instrument Manufacturing Co., Ltd., Shanghai, China) at a 4× magnification in the reflective mode.

### 2.10. Fat Distribution via a Confocal Laser Scanning Microscope (CLSM)

The fat distribution was measured according to the method of Shan et al. [23], with slight modifications. A 0.01% dyeing solution was prepared by dissolving 1 mg of Nile Red dye in 10 mL of acetone. The fried BBFNs were cut into slices (8 μm) at −20 °C by the refrigerated slicer (Cryotome E, Thermo Electron Co., Ltd., Shanghai, China), then fixed on the microscope slides. The slices were stained with the Nile Red dye solutions at 4 °C for 3 h and sealed with the anti-fluorescence quenching agent. Finally, the slides were placed on the carrier of a CLSM (OLYMPUS FV1200, Olympus Co., Ltd., Tokyo, Japan) for the observation. The CLSM parameters were: a 10 × magnification, scanning pixels of 1024 × 1024, a speed of 400 Hz, an excitation wavelength of 543 nm, and an emission wavelength of 638 to 768 nm.

### 2.11. Statistical Analysis

The analyses were performed in triplicate and the results were expressed as the means and standard deviation (SD). The data were processed and analyzed by applying Microsoft^®^ ExcelTM (Version 2007; Microsoft Corp., Redmont, WA, USA) and SPSS software (Version 17.0; SPSS Inc., Chicago, IL, USA). The analysis of variance among the groups was performed statistically by a one-way analysis of variance (ANOVA), and the Duncan test was used for the difference analysis of the mean values at the significance level (*p* < 0.05).

## 3. Results and Discussion

### 3.1. Fatty Acid Composition, Viscosity, and Dielectric Constant of Frying Oils

The dielectric constant provides information on the dielectric response of food to electromagnetic fields [33]. During deep-fat frying, polyunsaturated frying oils have a strong tendency to produce highly oxidized and polymerized compounds, which promotes an increase in polar substances, thereby directly affecting the distribution and strength of the electric field in frying oils, resulting in an increase in the dielectric constant [27,34,35,36]. In addition, polymers based on the dimer and trimer with branched chains were also formed because of the thermal oxidation and polymerization, resulting in an increase in the viscosity of the frying oils, which affect the adhesion capacity of the frying oils at the cooling stage [27].

The fatty acid composition of the fresh frying oils is shown in Table 1, and the viscosity and dielectric constant of the oils are depicted in Figure 1. The contents of the monounsaturated and polyunsaturated fatty acids of the oils are known to be positively correlated with the viscosity [21,26,37]. The unsaturated fatty acids content of the fresh frying oils was the highest for rapeseed oil, followed by soybean, rice bran, and palm oils. Soybean oil had the highest polyunsaturated fatty acid content among the four fresh oils, while palm oil exhibited the lowest level, which was generally consistent with the result of Zhang et al. [38]. The dielectric constant of the fresh frying oils was the greatest for palm oil, followed by rice bran, rapeseed, and soybean oils, a trend that was nearly opposite with the polyunsaturated fatty acid content. The change trends in the viscosity were generally consistent with fresh frying oils’ dielectric constant, except that the viscosity of the rice bran oil was higher than that of the palm oil (Figure 1). In addition, the preheating significantly increased the viscosity and dielectric constant of the frying oils (Figure 1). Similar results had been reported by Hwang et al. [39] and Yuan et al. [40]. The polymerization reactions that result from the degradation of the unsaturated fatty acids during the preheating lead to an increase in the frying oils’ viscosity [17,21].

### 3.2. Moisture and Fat Contents of Fried BBFNs

The effect of the frying oils on the moisture and fat contents of the fried BBFNs is shown in Table 2. The surface oil, penetrated surface oil, and total fat contents of fried BBFNs using preheated frying oils were significantly higher than those using fresh oils (*p* < 0.05), while the moisture content was significantly lower (*p* < 0.05). The penetrated surface oil content of the fried BBFNs using fresh oils increased with a decrease in the unsaturated fatty acid content (Table 1), while the total fat content followed the same trend, except that the content produced using soybean oil was lower than when employing rapeseed oil. These results indicated the significant influence of the oil quality on the fat absorption (*p* < 0.05).

During deep-fat frying, the moisture is transferred in the form of vapor and pores are formed, through which the frying oils may penetrate the foods [4,41]. The penetrated surface oil is mainly produced during the cooling stage. The frying oils are absorbed from the superficial oil layer as a result of the depressurization of batter-breaded foods. If there is more moisture evaporated, greater and larger pores on the surface of the crust would be formed, resulting in the more surface-adhered oil penetrating the core during the cooling stage. When the processes of moisture evaporation and oil penetration reach thermodynamic equilibrium, a high content of oil in the interior of the foods would be present [4,12].

The fat content of the fried BBFNs correlates well with the viscosity and dielectric constant of the oils employed, a result also observed by others [42]. For instance, among four fresh oils, the fat content was the highest for palm oil, the most viscous of the oils, while rapeseed oil, a much less viscous oil, produced the lowest penetrated surface oil content (Figure 1). A higher viscosity for the frying oils will allow more oil to adhere to the surface of fried foods after their removal from the fryer. The additional underlying factor is the formation of polymers and other polar compounds during deep-fat frying that possess a high surface activity, resulting in a decrease in the oil/water interfacial tension and the oil/crust contact angle, which increases the wetting of frying oils on the surface of fried foods and enhances the fat absorption [12,15]. It is likely that the formation of the polar and polymeric compounds led to the higher fat content of fried BBFNs using preheated frying oils compared to fresh oils [17].

### 3.3. Color Characteristics of Fried BBFNs

The effect of frying oils on the color characteristics of the fried BBFNs is shown in Table 2. For a given frying oil, the lightness (*L** value) of the fried BBFNs using fresh oils was higher than for preheated oils. However, the opposite trend was observed for the *a** and *b** values. The *L** value was the highest and the *b** and *a** values were the lowest for palm oil among the four fresh oils, and the lowest *L** value was obtained using fresh soybean oil. The golden color of fried foods is related to the Maillard reaction and caramelization. The moisture content levels of 10% to 15% are favorable to the Maillard reaction, with a lower moisture content forming a darker color [43,44]. The moisture contents of the crust using preheated oils were more favorable for the Maillard reaction compared to those of fresh oils (Table 2). Therefore, the lightness (*L**) of the preheated oils was smaller than for the fresh oils (Table 2). The color of the preheated oils was darker compared to the fresh oils, which could also affect the color of fried foods. Similar results were reported by Brannan et al. [14] and Gao et al. [25], in which oil degradation may increase the absorption of the visible light of oil, thereby leading to an increase in the darkness of oil, which affects the color of fried foods.

### 3.4. SEM Analysis

The effect of frying oils on the microstructural characteristics of the fried BBFNs is depicted through the SEM micrographs provided in Figure 2. The size of the pores and cracks in the crust and core obeyed the following order among the four fresh oils: palm oil > rice bran oil > soybean oil > rapeseed oil. The microstructure of the crust, the crust/core interface, and the core regions of the fried BBFNs were consistent with the analyses of the fat and moisture contents (Table 2). In addition, the pores and cracks of the fried BBFNs using preheated oils were larger than those using fresh oils. A higher content of polar compounds and polymers resulting from deep-fat frying hindered the uniform evaporation of the moisture. Therefore, non-uniform size pores and cracks were formed, which led to an increased fat absorption. Similar results were also observed by Lalam et al. [45].

### 3.5. Oil Transport Revealed by Optical Microscopy

The migration of frying oils into the fried BBFNs is observed by frying the BBFNs in oils that contained Sudan Red B dye (Figure 3). The Sudan Red coloring of the fried BBFNs resided in the crust rather than the core, with a small amount of red coloring at the interface between the crust and core. These results reveal that oil penetration mainly occurred around the crust and the interface between the crust and core [46,47]. Lalam et al. observed a 1–4 mm penetration layer of the oil in chicken nuggets after frying [45]. Bouchon et al. also proved that the oil penetration of fried foods was a surface phenomenon and determined, by using infrared thermography, that the penetration depth of oil into potato chips was only 300–400 μm [46]. In addition, employing the frying oils with a higher viscosity and lower surface tension would enhance the adhesion of the oil to the surfaces of fried foods, hindering the oil’s penetration through the crust pores into the core [48]. The largest dyeing area was observed for palm oil, followed by rice bran, soybean, and rapeseed oils. The order is consistent with the results for the penetrated surface oil (Table 2). Furthermore, the Sudan Red color intensity was higher for the preheated oils compared to the fresh oils, indicating that the content of the penetrated surface oil gradually increased more strongly for the preheated oils.

### 3.6. Fat distribution Analysis

The fat distribution of the fried BBFNs is observed by a CLSM (Figure 4), with the observed fluorescence intensity representing the distribution of the fat [23]. The fluorescence intensity was the highest for the fried BBFNs prepared using palm oil and changed with the oil type, and the fluorescence intensity vs. the fresh oil type relationship followed the same trend as observed for the Sudan Red color intensity and penetrated the surface oil content of the fried BBFNs (Table 2 and Figure 3). Interestingly, the fluorescence intensity of the fried BBFNs using the preheated oils was higher than for the fresh oils, which was consistent with the trends for the fat content (Table 2), confirming that the quality of the frying oils significantly affected the oil and moisture transformation of the BBFNs during deep-fat frying, hence leading to the difference in the fat content and quality attributes of the fried BBFNs.

## 4. Conclusions

The BBFNs that were fried using fresh oils with a high viscosity and dielectric constant or the preheated oils with high unsaturated fatty acids, possessed a high fat content and a darker color. The total fat contents of the fried BBFNs using fresh rapeseed oil, soybean oil, rice bran oil, and palm oil were 12.1%, 11.3%, 12.2%, and 14.2%, respectively. The unsaturated fatty acid content of the frying oils affected the oils’ viscosity and dielectric constant through the formation of polar compounds and polymers that reduced the interfacial tension between the frying oil/water system and the contact angle between the oil and the crust, thereby increasing the wettability of the frying oils during deep-fat frying, leading to the oil’s penetration of the fried BBFNs. These results could provide a scientific guidance for the manufacturing of fried BBFNs with a low fat content.

## Figures and Tables

**Figure 1 foods-11-03369-f001:**
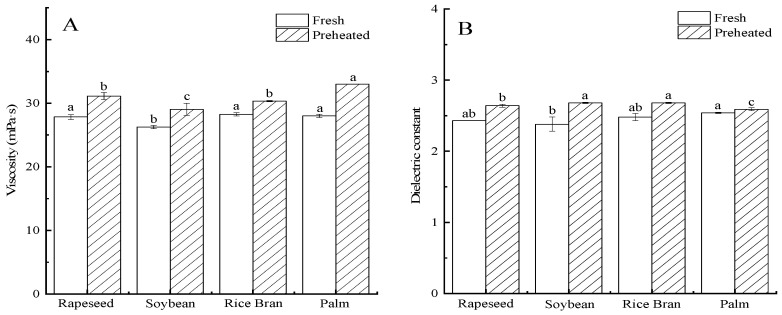
(**A**) Viscosity and (**B**) dielectric constant of fresh and preheated frying oils at 180 °C. Mean values listed in columns with different letters indicate statistically significant differences (*p* < 0.05).

**Figure 2 foods-11-03369-f002:**
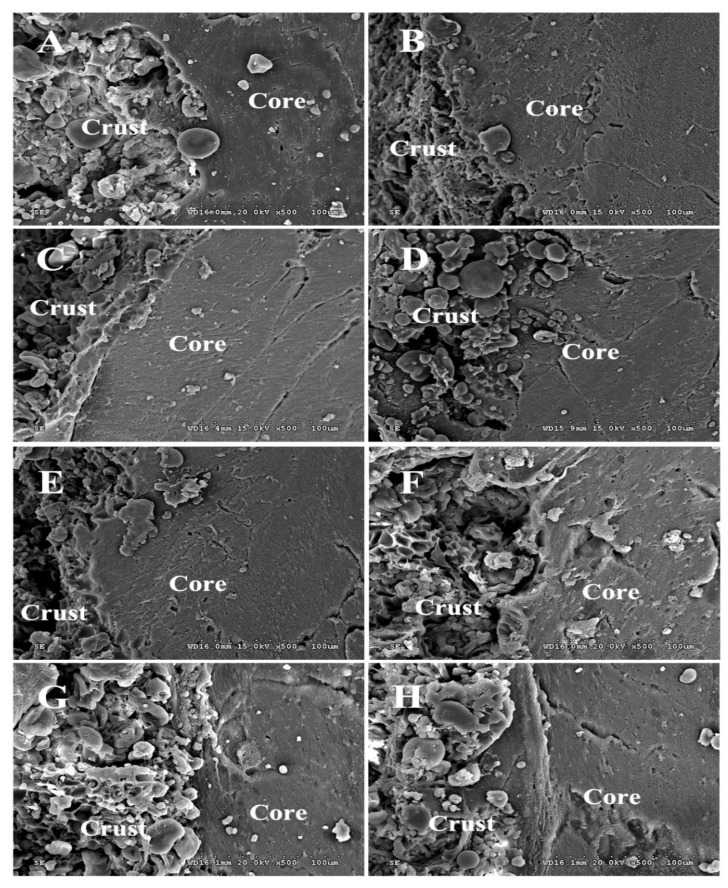
SEM micrographs of fried BBFNs using fresh and preheated frying oils. All the images were taken at 500× magnification. (**A**) Fresh rapeseed oil; (**B**) preheated rapeseed oil; (**C**) fresh soybean oil; (**D**) preheated soybean oil; (**E**) fresh rice bran oil; (**F**) preheated rice bran oil; (**G**) fresh palm oil; and (**H**) preheated palm oil.

**Figure 3 foods-11-03369-f003:**
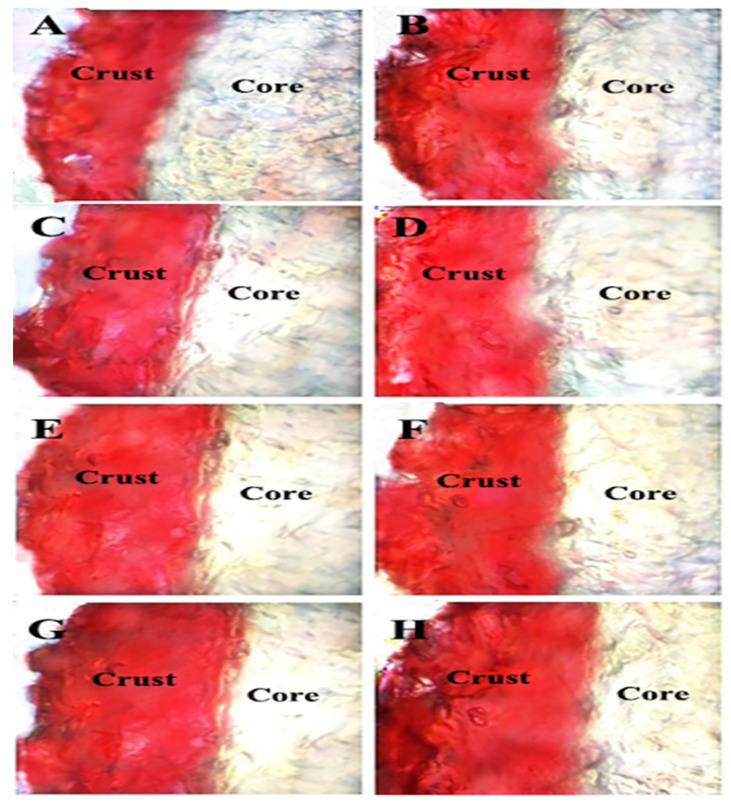
Optical microscopic images of fried BBFNs using Sudan Red B-containing oils. All of the images were taken at 4× magnification in reflective mode. (**A**) Fresh rapeseed oil; (**B**) preheated rapeseed oil; (**C**) fresh soybean oil; (**D**) preheated soybean oil; (**E**) fresh rice bran oil; (**F**) preheated rice bran oil; (**G**) fresh palm oil; and (**H**) preheated palm oil.

**Figure 4 foods-11-03369-f004:**
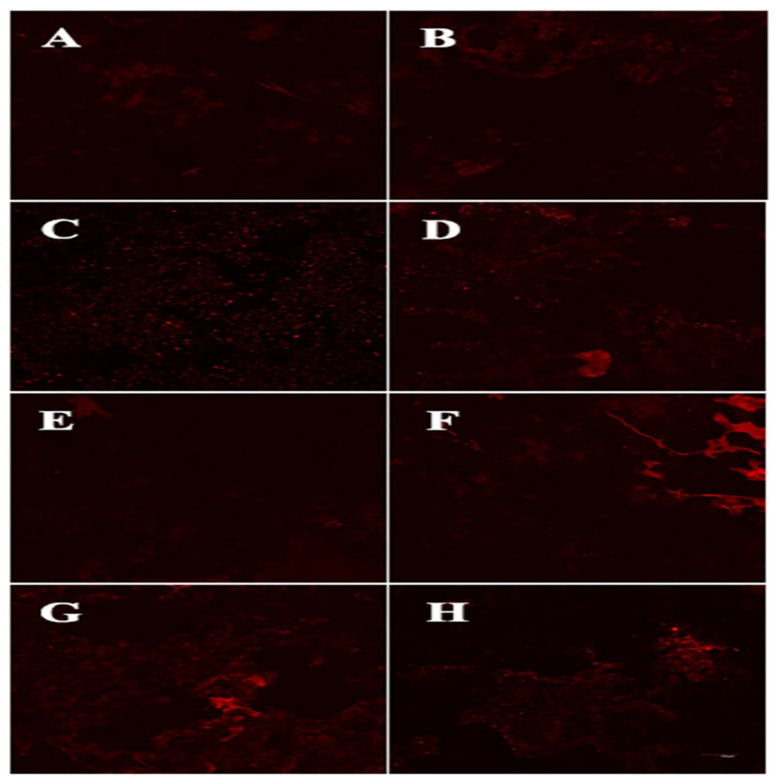
CLSM images of fried BBFNs using fresh and preheated frying oils. All of the images were taken at 10× magnification in fluorescence mode. (**A**) Fresh rapeseed oil; (**B**) preheated rapeseed oil; (**C**) fresh soybean oil; (**D**) preheated soybean oil; (**E**) fresh rice bran oil; (**F**) preheated rice bran oil; (**G**) fresh palm oil; and (**H**) preheated palm oil.

**Table 1 foods-11-03369-t001:** Fatty acid compositional estimation (relative percentage of the chromatographic areas) of fresh frying oils †.

Type of Fatty Acid	Rapeseed Oil	Soybean Oil	Rice Bran Oil	Palm Oil
Dodecanoic acid (C12:0)	-	-	-	0.2 ± 0.00
Myristic acid (C14:0)	-	0.1 ± 0.00	0.3 ± 0.00	0.9 ± 0.01
Palmitic acid (C16:0)	4.6 ± 0.03	11.6 ± 0.05	17.2 ± 0.00	39.3 ± 0.20
Palmitoleic acid (16:1)	0.2 ± 0.01	0.1 ± 0.00	-	0.2 ± 0.00
Heptadecanoic acid (C17:0)	-	0.1 ± 0.00	-	-
Octadecanoic acid (C18:0)	2.1 ± 0.01	5.0 ± 0.03	1.4 ± 0.01	4.7 ± 0.05
Oleic acid (C18:1)	60.4 ± 0.06	24.5 ± 0.01	40.7 ± 0.05	43.1 ± 0.07
Linoleic acid (C18:2)	20.8 ± 0.01	49.3 ± 0.02	37.3 ± 0.02	10.9 ± 0.12
Arachidic acid (C20:0)	0.7 ± 0.01	0.5 ± 0.02	0.6 ± 0.01	0.4 ± 0.01
Linolenic acid (C18:3)	9.5 ± 0.04	7.6 ± 0.53	1.5 ± 0.00	0.2 ± 0.05
Eicosenoic acid (C20:1)	1.2 ± 0.03	0.2 ± 0.00	0.6 ± 0.01	0.2 ± 0.01
Octadecadienoic acid (C18:2)	-	0.1 ± 0.00	0.1 ± 0.00	-
Heneicosanoic acid (C21:0)	-	-	0.4 ± 0.04	-
Docosanoic acid (C22:0)	0.5 ± 0.04	0.5 ± 0.02	-	-
Lignoceric acid (C24:0)	-	0.2 ± 0.00	-	-
Saturated fatty acid	7.8 ± 0.08 ^d^	17.9 ± 0.11 ^c^	19.9 ± 0.06 ^b^	45.4 ± 0.27 ^a^
Unsaturated fatty acid	92.2 ± 0.15 ^a^	81.8 ± 0.56 ^b^	80.1 ± 0.07 ^c^	54.6 ± 0.21 ^d^
Monounsaturated fatty acid	61.9 ± 0.10 ^a^	24.8 ± 0.01 ^d^	41.3 ± 0.05 ^c^	43.5 ± 0.08 ^b^
Polyunsaturated fatty acid	30.3 ± 0.05 ^c^	57.0 ± 0.55 ^a^	38.8 ± 0.02 ^b^	11.1 ± 0.13 ^d^

† Data are presented as mean value ± standard deviations from triplicate experiments (*n* = 3). “-” means that it was not detected. Mean values listed in columns with different letters indicate statistically significant differences (*p* < 0.05).

**Table 2 foods-11-03369-t002:** Moisture and fat contents and color characteristics of fried BBFNs using fresh and preheated oils †.

Oil Type	Preheating Duration (h)	Moisture Content (%)		Fat Content (%, dw)			Color Characteristics		
		Crust	Core	Surface Oil	Penetrated Surface Oil	Total	*L**	*a**	*b**
Rapeseed oil	0	15.3 ± 0.20 ^c^	68.4 ± 3.4 ^a^	5.0 ± 0.02 ^c^	6.3 ± 0.02 ^g^	12.1 ± 0.03 ^e^	48.2 ± 0.38 ^bc^	6.0 ± 0.67 ^e^	26.2 ± 0.30 ^d^
	10	13.4 ± 0.23 ^e^	62.2 ± 2.8 ^c^	5.5 ± 0.03 ^a^	6.5 ± 0.01 ^e^	12.8 ± 0.02 ^d^	46.5 ± 1.7 ^d^	6.7 ± 0.25 ^d^	27.8 ± 0.26 ^b^
Soybean oil	0	14.5 ± 0.33 ^d^	64.4 ± 1.1 ^b^	3.5 ± 0.04 ^g^	6.8 ± 0.02 ^f^	11.3 ± 0.03 ^f^	45.9 ± 1.3 ^de^	7.2 ± 0.23 ^c^	26.0 ± 0.29 ^d^
	10	13.2 ± 0.16 ^e^	58.2 ± 2.9 ^e^	5.0 ± 0.04 ^cd^	7.1 ± 0.04 ^de^	13.1 ± 0.03 ^c^	43.8 ± 1.5 ^f^	8.4 ± 0.32 ^a^	28.3 ± 0.53 ^ab^
Rice bran oil	0	17.1 ± 0.25 ^a^	64.2 ± 1.7 ^b^	4.3 ± 0.01 ^e^	7.1 ± 0.04 ^d^	12.2 ± 0.03 ^e^	47.0 ± 1.1 ^c^	7.7 ± 0.06 ^b^	27.5 ± 0.17 ^bc^
	10	14.2 ± 0.38 ^d^	61.1 ± 1.3 ^d^	5.4 ± 0.02 ^a^	7.9 ± 0.03 ^c^	14.1 ± 0.04 ^bd^	44.9 ± 1.2 ^e^	8.4 ± 0.40 ^a^	29.2 ± 0.08 ^a^
Palm oil	0	16.1 ± 0.27 ^b^	60.6 ± 2.7 ^d^	4.9 ± 0.04 ^d^	9.0 ± 0.04 ^b^	14.2 ± 0.02 ^b^	50.6 ± 1.2 ^a^	5.6 ± 0.17 ^f^	25.0 ± 0.85 ^e^
	10	12.2 ± 0.48 ^f^	57.5 ± 2.7 ^f^	5.3 ± 0.03 ^b^	9.4 ± 0.04 ^a^	15.3 ± 0.02 ^a^	48.1 ± 1.3 ^b^	7.7 ± 0.01 ^bc^	27.0 ± 0.19 ^c^

† Data are presented as mean value ± standard deviations from triplicate experiments (*n* = 3). Mean values listed in columns with different letters indicate statistically significant differences (*p* < 0.05).

## Data Availability

The data presented in this study are as described in the individual figures and tables.

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
