# Peer review of "Oil Penetration of Batter-Breaded Fish Nuggets during Deep-Fat Frying: Effect of Frying Oils"

_foods, 2022, doi:10.3390/foods11213369_

Round 1
Reviewer 1 Report
The paper presents results of a study on effects of frying oil quality on the quality of batter-breaded fish nuggets. Particularly, the penetration of oil in nuggets’ crust and core was examined, besides the crust color. Four vegetable oils were compared, both fresh and after a 10 h exposition to frying temperature.
The authors prove a relationship between oil viscosity and dielectric constant on one side and degree of oil penetration on the other side and indicate thus criteria for the choice of frying oil, as well as the importance of oil freshness.
The results are sufficiently presented in 4 figures and 2 tables. Still, I have a problem with understanding the Figure 4 with CLSM images. The significance of Figure 3 showing distribution of colored oil in the crust and core of nuggets is clear but for Figure 4 I would need answers to questions, which could be useful also to other readers: What part of nugget slices are depicted, crust, core, or both? Why are the shapes of red patterns in the eight images so different? The fluorescence intensity in Figure 4G seems to be higher than in Figure 4H.
Formally, the paper is very well prepared. Only a few small misprints should be corrected. I have found “Plam” instead of “Palm” under the last column in Figure 1, abbreviation Ep. instead of Eq. on lines 126 and 135, and LLYMPUS instead of OLYMPUS on line 231.
Author Response
The paper presents results of a study on effects of frying oil quality on the quality of batter-breaded fish nuggets. Particularly, the penetration of oil in nuggets’ crust and core was examined, besides the crust color. Four vegetable oils were compared, both fresh and after a 10 h exposition to frying temperature.
The authors prove a relationship between oil viscosity and dielectric constant on one side and degree of oil penetration on the other side and indicate thus criteria for the choice of frying oil, as well as the importance of oil freshness.
The results are sufficiently presented in 4 figures and 2 tables. Still, I have a problem with understanding the Figure 4 with CLSM images. The significance of Figure 3 showing distribution of colored oil in the crust and core of nuggets is clear but for Figure 4 I would need answers to questions, which could be useful also to other readers: What part of nugget slices are depicted, crust, core, or both? Why are the shapes of red patterns in the eight images so different? The fluorescence intensity in Figure 4G seems to be higher than in Figure 4H.
Response: The fluorescence mode of the microscope could reveal the oil distribution of the crust where the fluorescence intensity represented the oil content (Figure 4). During deep-fat frying, the rapid evaporation of water on the surface of batter-breaded foods may lead to surface hardening and the formation of pores, allowing the frying oil to penetrate the foods. Simultaneously, the frying oil also undergo a series of complex reactions, such as hydrolysis, polymerization, and oxidation, in which the triglycerides break down into polar mixtures (e.g., diglycerides, monoglycerides, free fatty acids, and glycerol), reducing the interfacial tension between the frying oil/water and the contact angle between the frying oil/foods, resulting in more frying oil to adhere to the crust surface [1, 2]. When the batter-breaded foods are removed from the hot oil after frying, the lowered temperature condensed water vapor inside the pores and cracks of the crust, creating a “vacuum effect” that draws the oil adhering to the surface of the crust into the pores and cracks, resulting in fat absorption of fried batter-breaded foods [3,4]. The oil absorption state of each fish nuggets is different, the shapes of red patterns were also different with the CLSM [5]. Although the fluorescence intensity in Figure 4G seems to be higher than in Figure 4H, the core of fish nuggets in Figure 4G were bigger than in Figure 4H (Table 2).
References:
[1] Sun, Y. N.; Zhang, M.; Fan, D. C. Effect of ultrasonic on deterioration of oil in microwave vacuum frying and prediction of frying oil quality based on low field nuclear magnetic resonance (LF-NMR). Ultrason Sonochem. 2019, 51, 77-89. https://doi.org/10.1016/j. ultsonch.2018.10.015.
[2] Xu, L. R.; Yang, F.; Li, X.; Zhao, C. W.; Jin, Q. Z.; Huang, J. H.; Wang, X. G. Kinetics of forming polar compounds in frying oils under frying practice of fast food restaurants. LWT-Food Sci. Technol. 2019, 115, 108307. https://doi.org/10. 1016/j. lwt.2019.108307.
[3] Brannan, R. G.; Mah, E.; Schott, M.; Yuan, S.; Casher, K. L.; Myers, A.; Herrick, C. Influence of ingredients that reduce oil absorption during immersion frying of battered and breaded foods. Eur. J. Lipid Sci. Technol. 2014, 116(3), 240-254. https://doi.org/10.1002/ejlt.201200308.
[4] Brannan, R. G.; Pettit, K. Reducing the oil content in coated and deep-fried chicken using whey protein. Lipid Technol. 2015, 27(6), 131-133. https://doi.org/ 10.1002/lite.201500022.
[5] Shan, J. H.; Chen, J. W.; Xie, D.; Xia, W. S.; Xu, W.; Xiong, Y. L. Effect of xanthan gum/soybean fiber ratio in the batter on oil absorption and quality attributes of fried breaded fish nuggets. J. Food Sci. 2018. 83(7), 1832-1838. https://doi. org/10. 1111/1750-3841.14199.
Formally, the paper is very well prepared. Only a few small misprints should be corrected. I have found “Plam” instead of “Palm” under the last column in Figure 1, abbreviation Ep. instead of Eq. on lines 126 and 135, and LLYMPUS instead of OLYMPUS on line 231.
Response: We have revised the manuscript extensively. We changed “Plam, Ep, and LLYMPUS” to “Palm, Eq, and OLYMPUS”, respectively. Please see the last column in Figure 1, line 126, line 135, and line 231.
I had the pleasure of reviewing your article. It deals with an important topic, but it is not new. Both the analysis and the oils used in the article are not new. What's new about the research?
Response: The highlights of the research are as following: (1) New report investigated the effect of frying oils on oil penetration of BBFNs. (2) Elevated unsaturated fatty acids content increased fat absorption of fried BBFNs. (3) Oil penetration of fried BBFNs correlated with freshness of frying oils.
Reviewer 2 Report
Dear Authors,
I had the pleasure of reviewing your article. It deals with an important topic, but it is not new. Both the analysis and the oils used in the article are not new. What's new about the research?
Other comments for the work:
- please remove any unnecessary spaces in the text
- fatty acid composition analysis: please indicate how the acid composition is given, whether quantitatively or qualitatively. Was the injection really 1 microliter and not 0.8? I think it would be better to quantify the composition. Because I will be losing all acids, only some in greater amounts, others in less.
- Table 1, please put it in the right place in the text as soon as it is quoted.
- Line 323, please specify what these similar values ​​were; the same applies to other analyzes
- Line 328, what was the pore size, the smaller the bigger ones say nothing, the given dimensions would make it possible to compare the obtained results better.
Author Response
Other comments for the work:
- please remove any unnecessary spaces in the text
Response: We have revised the manuscript extensively. Please see the manuscript.
- fatty acid composition analysis: please indicate how the acid composition is given, whether quantitatively or qualitatively. Was the injection really 1 microliter and not 0.8? I think it would be better to quantify the composition. Because I will be losing all acids, only some in greater amounts, others in less.
Response: Fatty acid composition was expressed as percentage of the total peak area of all the fatty acids in frying oils. The injection volume was 1.0 μL with a split ratio of 50:1. Please see line 112 and lines 117-118.
- Table 1, please put it in the right place in the text as soon as it is quoted.
Response: We have revised the manuscript extensively. Please see manuscript.
- Line 323, please specify what these similar values were; the same applies to other analyzes
Response: We have revised the manuscript extensively. Please see lines 322-323.
- Line 328, what was the pore size, the smaller the bigger ones say nothing, the given dimensions would make it possible to compare the obtained results better.
Response: We have revised the manuscript extensively. Please see line 326, line 330, and line 333.